# Sustainable Diesel from Pyrolysis of Unsaturated Fatty Acid Basic Soaps: The Effect of Temperature on Yield and Product Composition

**DOI:** 10.3390/molecules27030667

**Published:** 2022-01-20

**Authors:** Endar Puspawiningtiyas, Oki Muraza, Hary Devianto, Meiti Pratiwi, Tirto Prakoso, Usamah Zaki, Lidya Elizabeth, Tatang H. Soerawidjaja, Yohanes Andre Situmorang, Antonius Indarto

**Affiliations:** 1Department of Chemical Engineering, Institut Teknologi Bandung, Ganesha Street No. 10, Bandung 40132, Indonesia; endartiyas@yahoo.com (E.P.); hardev@che.itb.ac.id (H.D.); mei@che.itb.ac.id (M.P.); subagjo@che.itb.ac.id (S.); tirto@che.itb.ac.id (T.P.); krisnawanj@gmail.com (K.); usamah.zaki96@gmail.com (U.Z.); thsoerawidjaja@gmail.com (T.H.S.); 2Department of Chemical Engineering, Universitas Muhammadiyah Purwokerto, Purwokerto 53182, Indonesia; 3Research & Technology Innovation, Pertamina, Sopo Del Building, 51st Fl. Jl. Mega Kuningan Barat, Jakarta Pusat 12950, Indonesia; oki.muraza@pertamina.com; 4Departement of Bioenergy Engineering and Chemurgy, Institut Teknologi Bandung, Jalan Let. Jen. Purn. Dr. (HC). Mashudi No. 1 Sumedang, Kota Bandung 45363, Indonesia; yohanes.andrest@gmail.com; 5Department of Chemical Engineering, Politeknik Negeri Bandung, Gegerkalong Hilir Street, Bandung 40559, Indonesia; lidya.elizbth@gmail.com

**Keywords:** pyrolysis, unsaturated fatty acid, basic soap, metal hydroxide, biohydrocarbon

## Abstract

The production of sustainable diesel without hydrogen addition remains a challenge for low-cost fuel production. In this work, the pyrolysis of unsaturated fatty acid (UFA) basic soaps was studied for the production sustainable diesel (bio-hydrocarbons). UFAs were obtained from palm fatty acids distillate (PFAD), which was purified by the fractional crystallization method. Metal hydroxides were used to make basic soap composed of a Ca, Mg, and Zn mixture with particular composition. The pyrolysis reactions were carried out in a batch reactor at atmospheric pressure and various temperatures from 375 to 475 °C. The liquid products were obtained with the best yield (58.35%) at 425 °C and yield of diesel fraction 53.4%. The fatty acids were not detected in the pyrolysis liquid product. The gas product consisted of carbon dioxide and methane. The liquid products were a mixture of hydrocarbon with carbon chains in the range of C_7_ and C_20_ containing n-alkane, alkene, and iso-alkane.

## 1. Introduction

Increased energy demand and issues related to environmental concerns constitute a strong reason many countries have chosen to use alternative and renewable energy technologies. Biohydrocarbon or liquid hydrocarbon are renewable fuels derived from any material originating from biological matters [1], including biomass [2], lignocellulose [3], triglycerides, or fatty acids [4,5]. There are many reasons for the strong interest in biofuels, among others their easy availability from common biomass. Biofuels represent a carbon dioxide cycle in combustion. They have considerable environmentally friendly potential, and they are biodegradable and contribute to sustainability [6]. Nowadays, the main processes used to obtain biofuels from vegetable oils are transesterification and thermal cracking (pyrolysis) or thermal-catalytic cracking (catalytic pyrolysis) [7]. Biodiesel, a promising biofuel, is made from renewable biological sources, such as vegetable oils and animal fats, by chemically reacting oil or fat with an alcohol (transesterification) in the presence of a homogeneous and heterogeneous catalysts [8]. Biodiesel is renewable, biodegradable, nontoxic, and produces low emission during combustion [9]. However, there are some disadvantages of biodiesel, as listed by Baladincz et al. [10], such as (a) high unsaturated bond content of fatty acids (causing bad thermal oxidation, and thus instability); (b) relatively high hygroscopic property; (c) reactive OH-group cause corrosion of metals; (d) relatively lower calorific value which causes high fuel consumption, and (e) unfavorable cold flow properties. According to Solymosi et al. [11], biodiesel, as one of the first generation biofuels, remains proportionally limited when used as a fuel due to the properties of this compound. According to [12], biodiesel contains about 10% oxygen by weight.

Among thermochemical processes, pyrolysis has received increased interest since the process conditions can be optimized to maximize the production of chars, liquids, or gases [13]. Thermal pyrolysis operates at 700–1000 °C and produces mostly gaseous products containing straight-chain hydrocarbon fuels, according to Araújoet et al. [7]. The pyrolysis of fatty acids or triglycerides into biofuel can be achieved from different sources of fatty acids or triglycerides, including soybean (*Glycine max*) by Junming et al. [14], sunflower by Yigezu and Muthukumar [15], *jatropha* oil by Biswas et al. [16], waste shipping oil by Wan Mahari [17], trilaurin and trimyristin by Chiappero et al. [18], cotton seed oil by Li et al. [19], and Canola by Idem et al. [20].

According to Demirbaş [21], the soap obtained by the saponification of vegetable oils can produce hydrocarbon with upper yield at higher temperatures. The saponification of triglycerides prior to the pyrolysis process is proposed as a step towards obtaining compositions of liquid product similar to diesel fuel. The liquid product yield of soap pyrolysis was higher than triglyceride pyrolysis. In addition, it contains less oxygenated compounds according to Lappi and Alén [22]. There are advantages of soap pyrolysis products rather than triglycerides, including lower acid value [23], lack of polymerization, less sludge formation, and lower viscosity, according to Hiebert [24]. Furthermore, it was more stable and demonstrated less tendency to form wax, as reported by Joonwichien et al. [25].

Most of the previous researches used soap pyrolysis to produce hydrocarbon according to Kaisha [26]. That report considered the pyrolysis reaction mechanism of stoichiometric and basic sodium soap. It showed that, at pyrolysis temperature, stoichiometric sodium soap produced not only short chain hydrocarbon, but also ketone and aldehyde, whereas the pyrolysis of basic sodium soap produced only hydrocarbon. This mechanism was supported by Hites and Biemann [27], who reported that the liquid product from pyrolysis of calcium decanoic stoichiometric soap usually contained ketone compounds and carbonyl group. The compounds disappeared at the upper temperature of 500 °C. This showed that the basic is better than the stoichiometric soap to produce a liquid product free from oxygenated compounds. Other reviews within the same field include those by Hiebert et al. [24] and Kufeld et al. [28]. Most of the works used single metal, while there are still a few references that reported the use of two or three metals to obtain hydrocarbon via the pyrolysis of basic soap. In addition, the decarboxylation reaction occurred when divalent metal base soaps, such as calcium and magnesium, were heated without air under atmospheric pressure, in the temperature range 200–400 °C, producing hydrocarbon, according to Markley et al. [29].

The production of hydrocarbons via pyrolysis of the basic soap was investigated in this research. Figure 1 shows the technology of basic soap pyrolysis in the scope of the biofuel production from vegetables oils. The figure shows oils/fats or fatty acids as the raw material for making hydrocarbon (diesel or gasoline). The first step is oils or fatty acids saponification by a metal hydroxide/oxide, which produces metal soap. When the raw material consists of fatty acids, it produces water as by-product, while glycerol is produced when using oil/fat. Then, diesel type hydrocarbons (green-diese) are obtained by the decarboxylation of metal soap, and biogasoline by pyrolysis. Metal hydroxide/oxide as a by-product of decarboxylation and pyrolysis is recycled to saponification material. The aim of this research was to investigate the effect of pyrolysis temperature of the basic soap of UFA on liquid product composition. In addition, a calcium-magnesium-zinc hydroxide combination to prepare the basic soap was designed.

## 2. Methodology

### 2.1. Materials

The unsaturated fatty acids were obtained from palm fatty acids distillate (PFAD) separation by the fractional crystallization method [30]. PFAD used in this research was received from PT Garmex Biofuel. Calcium chloride dihydrate (CaCl_2_^●^H_2_O), magnesium chloride hexahydrate (MgCl_2_^●^6H_2_O), and zinc chloride tetrahydrate (ZnCl_2_^●^4H_2_O) were supplied by Merck and were used to prepare mix-metal hydroxide by co-precipitation with sodium hydroxide [31]. The Ca-Mg-Zn soap was obtained by a saponification reaction of unsaturated fatty acids with mixed metal hydroxide [32,33,34].

### 2.2. Preparation of Unsaturated Fatty Acids

The PFAD (100 g) was dissolved into 200 g of solvent A comprising 85%-w acetonitrile with 15%-w of water at 55 °C. The solution was cooled to room temperature while stirring. The precipitate was formed, filtered by vacuum funnel, and washed twice using acetonitrile solution. Furthermore, a precipitate which consisted of an unsaturated fatty acid was weighed and stored for iodine and saponification number analysis.

### 2.3. Preparation of Mixed Metal Hydroxides

The mixed metal hydroxide was prepared via the coprecipitation method of sodium hydroxide solution with high carbonates/nitrates base solution. The first burette (A) contains 50 mL of 0.1 mol (Ca + Mg + Zn) nitrate combination solution and the other burette (B) 50 mL of 0.2 mol NaOH solution. The ratios of Ca to Ca/Mg/Zn mixture were chosen in the range of 15–85% by weight. The three metal inclusion points were expressed as 0.15 μ and 0.50 μ (mole ratios). The co-precipitation is controlled by the rate of drop burette B to keep the pH of slurry at 9.6. After continuous stirring for 30 min, it was then separated by filtering and washing with water. Furthermore, the slurry was dried at 105 °C for 12 h.

### 2.4. Preparation of Basic Soap

The UFA (0.1 mol) and 0.05 mol of mixed metal hydroxide were fed into a saponification reactor while mixed. Then, the mixture was heated to 40–45 °C before adding 4 mL of water and 0.18 g of formic acid 98–100% purity. The mixing was completed in at least 30 min. Then, the basic soap was formed and dried at 60 °C.

### 2.5. Pyrolysis of Basic Soap

The pyrolysis of the basic soap was performed in a 100 mL glass batch reactor equipped with two thermocouples for the liquid and vapor phases, a glass condenser, and a liquid product collector. A total of 10 g of the basic soap was added to the reactor. N_2_ gas flowed into the reactor for about 10 min to remove oxygen before it was heated to the specified temperature. The temperature variables were 375, 400, 425, and 450 °C under atmospheric pressure for 3 h. The gas product formed, condensed, and was collected in a liquid product collector. When the liquid product was formed, the uncondensed gas product was accumulated in a gas catcher which was installed in the liquid product collector. The pyrolysis liquid products were analyzed by GC-FID [1], while GC-TCD was adopted for the gas product. The pyrolysis apparatus was setup according to previous research [35].

### 2.6. Measurement Values and Units

The pyrolysis of mixed metal basic soap resulted in a series of products observed as periodic sets of peaks when analyzed by GC-FID. The GC-FID Shimadzu 2010 was equipped with a capillary column (rtx-1) with flame ionization detector and dimensions of 30 × 0.25 × 0.25 μm. Sample (1 microliter) was injected into the GC by helium (the carrier gas) flow rate was 42.9 mL/min. The detector and injector temperature were 340 °C. The following chromatographic temperature program was used for analysis: 40 °C (at first)–300 °C (5 °C/min)–340 °C (1 °C/min, constant 45 min). The typical composition of UFA was obtained by PFAD separation and used as renewable feedstock in the saponification reaction analyzed by GC-MS.

The gas products of basic soap pyrolysis were analyzed by GC TCD Shimadzu type GC-8A. The sample was injected into the GC with argon (the carrier gas), the flow rate was 1:3.5 kg/cm^3^, and carrier gas flow pressure was 2:3.5 kg/cm^2^. The detector and injector temperature were 70 °C and 50 °C. The qualitative information of oxygen functional groups or oxygenates in the liquid bio-hydrocarbon was analyzed using a Ferrox paper test [36].

## 3. Results and Discussion

### 3.1. Unsaturated Fatty Acids and Mixed Metal Basic Soap Analysis

The typical composition of UFA obtained by PFAD separation, which was used as renewable feedstock in the saponification reaction analyzed by GC-MS, is presented in Table 1. The result shows the total UFA content from PFAD separation of 9-octadecenoic acid (59.9%-wt) with 9,12-octadecadienoic acid (5.4%-wt). Furthermore, based on data of PFAD shown Table 1, the UFA composition in the feed of PFAD 45.48%-wt (C18:1 40.3%-wt and C18:2 5.2%-wt) means that UFA composition increased 19.9%-wt. This indicated that separation of UFA from PFAD had gone well. The mixed metal basic soap of UFA from PFAD presented a saponifiable hydroxide value of 49%, indicating a good soap from 50% as target. The basicity of soap was ascertained by detecting the O-H group through FTIR analysis. Figure 2 illustrates the FT-IR analysis of M-mix basic soap, showing the detection group O-H, C-H, C=C, dan C-C at wavenumber 3200–3700 cm^−1^, 2800 –3000 cm^−1^, 1500–1645 cm^−1^, and 1300–1500 cm^−1^, respectively [37]. The O-H groups indicate that it was basic soap and qualified as pyrolysis feed for biohydrocarbon production. The presence of groups detected in the soap matched the chemical formula of the basic soap: μCa(OH)_2_^●^(1-µ)Mg(OH)_2_^●^Zn(OOCC_17_H_33_)_2_.

### 3.2. Identification of Product in the Liquid Fraction

Table 2 presents material balance of biohydrocarbon products from the pyrolysis of M-mix basic soap, where liquid biohydrocarbon yield at 425 °C was 58.4%-wt, higher than 375 °C (51.6%-wt), 400 °C (55.9%-wt), and 450 °C (50.1%-wt). The phenomena observed could be explained at temperature pyrolysis above 425 °C, with the potential to accelerate cracking of high molecular weight hydrocarbon into low molecular weight hydrocarbon. The temperature trend was also reported by Neonufa et al. [38] in the decarboxylation of mixture metal stearin soap. Table 2 shows that the yield of liquid product increased with pyrolysis temperatures enhanced up to 425 °C and decreased by about 8%-wt at 450 °C. This can be explained as follows: at elevated temperature from 425 °C to 450 °C, the cracking of biohydrocarbon chains already obtained as a decarboxylation product has been expected. The result of this cracking is a shorter chain biohydrocarbon which could be condensed or not (as CH_4_). This is supported by Figure 6, which shows that at 450 °C, the biohydrocarbon composition of the short carbon chain is highest compared to other temperatures. The result of this study is also supported by previous studies of [27] which reported that, at 450 °C, all stoichiometric calcium decanoate soap obtained complete decarboxylation. Then, the soap begins to crack into shorter hydrocarbons chains and oxygenate compounds (aldehydes or ketone). The difference is that the liquid product in this study does not contain oxygenate compounds. This caused the pyrolysis feed to be basic soap, not stoichiometric.

The solid product of basic soap pyrolysis is a mixture of metal carbonates. The formation of the product indicates the phenomenon of the release of carbon dioxide (-COO) in basic soap. The pyrolysis reaction of basic soap is shown in Equation (1). Table 2 shows that the yield of solid product increases at 375–425 °C. This implies that the rate of -COO group removal increases with temperature. Furthermore, the yield of solid product decreases (13.9%-wt) at 450 °C. The plausible reason for this phenomenon is that at a temperature of 450 °C some of the metal carbonates have decomposed into metal oxide.
M_mix_(OH)(OOCH-C_7_H_15_-CH=CH-C_8_H_17_)→2C_17_H_34_ + M_mix_CO_3_(1)

Figure 3 shows that there are similarities in the peak chromatogram trend in all temperature variations. The pyrolysis of mixed metal basic soap produced a hydrocarbon-like mixture containing a carbon chain length between C_7_ and C_20_. The graph of the pyrolysis liquid product component (Figure 3) shows that liquid products of basic soap pyrolysis contain n-alkane, iso-alkane, and 1-alkene, while the most dominant component is iso alkane. This wide range of isoalkanes shows that, in addition to the decarboxylation reaction and cracking, the hydrogenation reaction was possibly followed by isomerization. A similar observation was made concerning oleic acid pyrolysis [39]. The identification of fatty acids in the liquid product of basic soap pyrolysis has been carried out at the lowest and highest temperature (Table 3). The analysis is in accordance with the AOCS CD 3D-63 method. The measurement shows that the liquid products of mixed metal basic soap pyrolysis have a low acid value, which means that the hydrocarbon-like fuel mixtures can be classified as biofuels. In addition, the ferox analysis for liquid product has also been performed, where an unidentified oxygenated compound was qualitatively observed. These results are different from the previous soap pyrolysis study [40]. The report shows that the liquid pyrolysis products of Ca, Mg, and Zn oleate soaps were detected in a carbonyl group (C=O). This led to the presence of fatty acids in the liquid pyrolysis product. This difference explains that the type of reactor and the feeding method used differ with this study. This study used a glass reactor with a batch feeding method, while previous studies adopted a stainless steel reactor and a semi-continuous feeding method.

Furthermore, the result in this study is also different from oleic acid pyrolysis research [39]. The report explained that the pyrolysis of oleic acids at 350–500 °C identified a series of fatty acids in the liquid product of oleic acid pyrolysis. This shows that the use of metals in this study succeeded in removing the -COO group in fatty acids. Furthermore, it proved that soap pyrolysis better than fatty acid pyrolysis to produce sustainable diesel.

### 3.3. Effects of Temperature on the Composition of Alkane and Alkene Compounds in the Liquid Products

Figure 4 shows the alkane and alkenes composition of the liquid product from mixed-metal basic soap pyrolysis at different temperatures and different Ca/(Ca + Mg + Zn) metal ratios. The yield of alkanes in liquid products reached a maximum (25.9%) when the pyrolysis was carried out at temperature of 400 °C. According to [41], when cracking occurs, a single bond on the side of the double bond will be cracked into two alkene compounds. Furthermore, alkenes undergo hydrogenation to obtain alkanes. Based on this mechanism, the yield of alkenes should be higher than alkanes, but not with this research. In this study, the amount of alkanes and alkenes at each temperature variation is almost similar. According to Neonufa [38], alkenes in biohydrocarbons are caused by other reactions than decarboxylation, namely the dehydration reaction of M-mix(oleic)(OH) soap. The reaction produces dehydrated, partially M-mix(oleic)(OH) soap and water. Then, the decarboxylation of the soap produces a mixture of alkane and alkene. When the amount of alkanes and alkenes is almost the same, the dehydration of basic soap coincides with the decarboxylation reaction. The next step is the hydrogenation of alkenes. The presence of hydrogen may be obtained from alkene decomposition with carbon as another product [39]. This was proven by the formation of black solid carbon after the pyrolysis. At pyrolysis temperatures from 375 to 400 °C, the selectivity pathway towards alkane formation occurred. Meanwhile, at temperatures from 425 °C to 450 °C, the selectivity pathway towards the formation of alkene occurred. However, the selectivity trend is insignificant.

The data from Figure 4 show that alkene yields are almost similar for all temperature variations, averaging around 17.5%. This fact is due to the batch feeding in this study, which causes the basic soap to heat slowly before reaching pyrolysis temperature. Then, the basic soap undergoes a relatively long decarboxylation time. According to Neonufa [38], high levels of alkenes in biohydrocarbons are caused by other reactions than decarboxylation, namely the dehydration reaction of M-mix(oleic)(OH) soap. The reaction produces dehydrated, partially M-mix(oleic)(OH) soap and water. Then, the decarboxylation of the soap produces mixture of alkane and alkene. Finally, the high level of alkenes are due to the dehydration of basic soap that occurs before or concurrently with decarboxylation. This result is supported by previous research [40] considering the effect of metal type on basic soap pyrolysis. The report showed that the semi-continuous pyrolysis of oleic Ca, Mg, and Zn basic soap, respectively, produce biohydrocarbons containing fewer alkene compounds. This indicated that, in pyrolysis with semi-continuous feeding, basic soap is more able to retain hydroxide than batch feeding before pyrolysis temperature.

### 3.4. Effect of Temperature on Iso-Alkane Compounds in the Liquid Products

The effect of temperature on iso-alkane compounds from liquid products is shown in Figure 5. At all temperatures, the liquid pyrolysis products had a higher amount of saturated products (n-alkane and iso-alkane) as compared to unsaturated products (alkene), where possible hydrogenation occurred after the pyrolysis. A similar observation was reported by Asomaning et al. [39] with n-alkanes as the main saturated product, whereas in this study the main product was iso-alkane (Figure 5 and Figure 6). The observed difference was attributed to Zn used as a hydroxide mixture as a transition metal that could play a role in the isomerization of hydrocarbon as reported by Fontaine et al. [42]. The amount of iso-alkane at all temperatures was almost same. A reasonable explanation for this was related to the M-mix soap rupturing at temperatures below 400 °C, forming carbonates and long chain hydrocarbon. Furthermore, the products were hydrogenated and partially isomerized. Isomerization products persist up to 450 °C. According to results of GC-TCD analysis of M-Mix soap pyrolysis gas products (Table 4), carbon dioxide and methane were also detected in the gas product. The previous study has described the formation of iso-alkane through decarboxylation and hydrogen transfer, respectively, from fatty acids. Based on the results shown in Figure 5, it can be noted that to produce a liquid product with the highest isomer product yield, the pyrolysis temperature must be 425 °C as catalyst selectivity to isomerization products was influenced by the temperature.

The mixed metal basic soap has perhaps the most selective catalyst activity due to the isomerization at temperature of 425 °C. According to Masudi et al. [43], there are two methods to induce branch formation in hydrocarbons, namely (i) direct isomerization from saturated carbon and (ii) isomerization from unsaturated carbon, which is followed by hydrogenation. Figure 6 shows the hydrocarbons with different carbon numbers from the pyrolysis of mixed metal basic soap at various temperatures with a Ca metal ratio of 0.15µ. The figure shows that iso alkanes are the dominant component of liquid biohydrocarbon products in almost all products with different numbers of carbon in all temperature variations. This component was the most dominant in C_16_ and C_17_. This shows that the pyrolysis of basic soaps began with
decarboxylation. Referring to Table 1, the components of basic soap feed were dominated by oleic acids (9-octadecenoic acids), 60%. Then, the decarboxylation of basic soap formed heptadecane (Equation (2)). After that, the hydrogenation of heptadecene formed heptadecane (Equation (3)), and heptadecane was isomerized (Equation (4)). This mechanism shows that the formation of iso alkane components in this study was attributed to the direct isomerization from saturated carbon.


      C_8_H_17_-CH=CH-C_7_H_14_-CO-O-M_mixed_-OH → C_8_H_17_-CH=CH-C_7_H_15_ + M_mixed_CO_3_Metal-oleic basic soap → Heptadecene + Mixed-metal carbonate(2)
     C_8_H_17_-CH=CH-C_7_H_15_ + H_2_
→ C_8_H_17_-CH_2_-CH_2_-C_7_H_15_Heptadecene + Hydrogen → Heptadecane(3)
C_8_H_17_-CH_2_-CH_2_-C_7_H_15_
→ i-C_17_H_36_   Heptadecane → heptadecane isomers(4)


### 3.5. Effects of Temperature on Yield of Gasoline, Avture and Diesel Hydrocarbon Fractions

The chosen operating parameters of pyrolysis, such as temperature, affect the quality of the liquid product, which mainly consisted of the fraction of hydrocarbon consisting of light fraction or C_7_–C_11_, moderate fraction or C_12_–C_15_, and heavy fraction or C_16_–C_20_. Figure 7 shows the effect of temperature on the yield of light, moderate, and heavy fractions of liquid product from the pyrolysis of mixed-metal basic soap. Based on the results shown in Figure 7, the yield of light fraction liquid product has an average around 20%. Kaisha et al. [26] studied soap pyrolysis and found that the yield of gasoline or light fraction in the liquid product reached 25%. This indicated that the pyrolysis products obtained in this study were in line with those reported by Kaisha et al. Nevertheless, the results of moderate and heavy fractions in this study were different than those reported previously. In this study, the moderate or jet fuel (avtur) fraction was 35%, while Kaisha et al. reported 55–60%. Moreover, the heavy or diesel fraction in this study was 45%, while they reported in the range of 15–20%. Other studies previously only reported the types of fractions in liquid products without a quantitative amount. This means that the yield of diesel fraction in this study is higher than previous research.

Figure 7 shows that the highest yield of light fraction (C_7_–C_11_)/gasoline was obtained at 375 °C (27.1%). This result is not supported by previous research where the optimal fraction was obtained when the temperature pyrolysis was higher. In addition, Asomaning [39] reported that the best raw material to produce biogasoline by fatty acid pyrolysis is unsaturated fatty acid. This could be explained as fatty acids cracked at pyrolysis temperature to form a shorter hydrocarbon chain, while at decarboxylation temperature, fatty acids removed the carboxyl group. This statement is supported by Shim [44]. The report shows that the deoxygenation of oleic acids in a hydrogen environment at 300 °C produced C_9_–C_17_ hydrocarbon compounds (diesel fuel range). The possible reason for the gasoline yield in this research, reached at 375 °C, is the long chain hydrocarbon from decarboxylation undergoing further decomposition [38]. This occurs due to the long time of pyrolysis, so that the basic soap evaporates and condenses back into reactor with high frequency. When this occurs continuously, the cracking also continues.

The data from Figure 7 indicate that the highest yield of avture fraction (C_12_–C_15_) is reached at 375 °C (40.7%). This is supported by Hites and Biemann [27] as the metal soap cracked at above 400 °C produces shorter hydrocarbon compounds as the dominant product. Decarboxylation of metal soap occurs before metal soap cracking. The process removes the carboxyl group (-COO) from unsaturated fatty acids. The highest diesel fraction (C_16_–C_19_) was reached at 425 °C (53.4%). This shows that the long chain hydrocarbon remains in the pyrolysis liquid product above 400 °C. This result is not supported by previous research. This research showed that the best raw material to produce diesel fraction fuel by pyrolysis of fatty acids is saturated fatty acids [45]. A possible reason to explain this phenomenon is that the system used in this research is slow pyrolysis. In this pyrolysis, when the pyrolysis temperature has not reached 400 °C, there is a liquid product drop in the liquid product flask. The liquid product contains a lot of long chain hydrocarbons mixed with liquid products that drop above 400 °C. However, other researchers report that the pyrolysis of unsaturated fatty acids (oleic acids) at 450 °C also produced diesel-range hydrocarbon, although with a different catalyst [46].

## 4. Conclusions

Mixed metal basic soaps were converted to liquid biohydrocarbon by pyrolysis. The best yield (58.35%) was obtained at 425 °C with the higher diesel fraction of 53.4%. No fatty acids were detected in the liquid biohydrocarbon. The pyrolysis of basic soap forces the metal to attract the –COO group on the soap to remove almost all of the group. This is the basic difference between basic soap pyrolysis and fatty acids. This proves that basic soap was a better feed for pyrolysis to produce sustainable diesel as compared to fatty acid feeds. The content of liquid product of basic soap pyrolysis is dominated by diesel range biohydrocarbons (C_16_–C_19_). The product consists of normal alkane, alkene, and the various iso-alkane products.

## Figures and Tables

**Figure 1 molecules-27-00667-f001:**
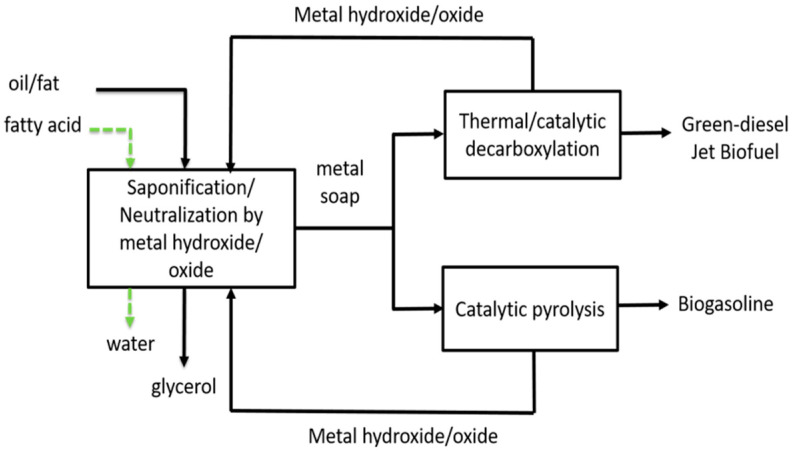
Flow diagram of biofuel production from vegetable oil/fatty acids.

**Figure 2 molecules-27-00667-f002:**
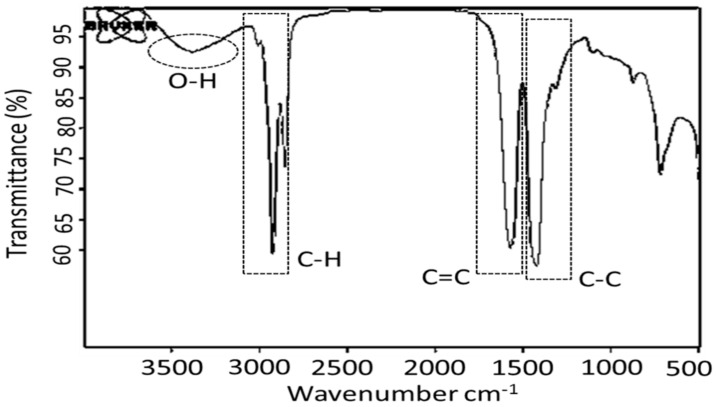
FT-IR of mixed-metal basic soap of unsaturated fatty acid.

**Figure 3 molecules-27-00667-f003:**
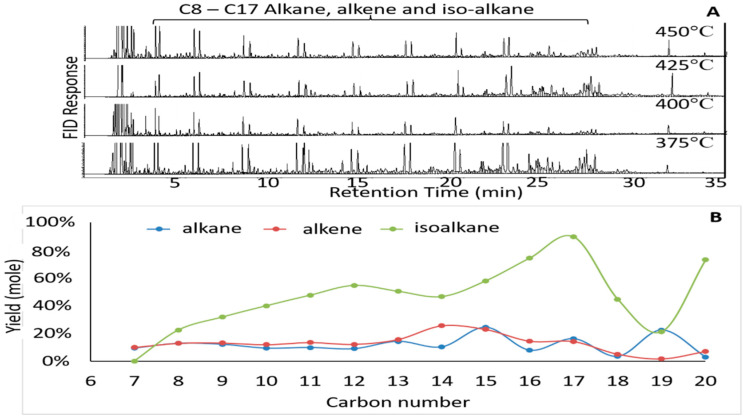
GC-FID chromatogram at different temperature (**A**) and hydrocarbon component of liquid products from pyrolysis of unsaturated fatty acid (**B**).

**Figure 4 molecules-27-00667-f004:**
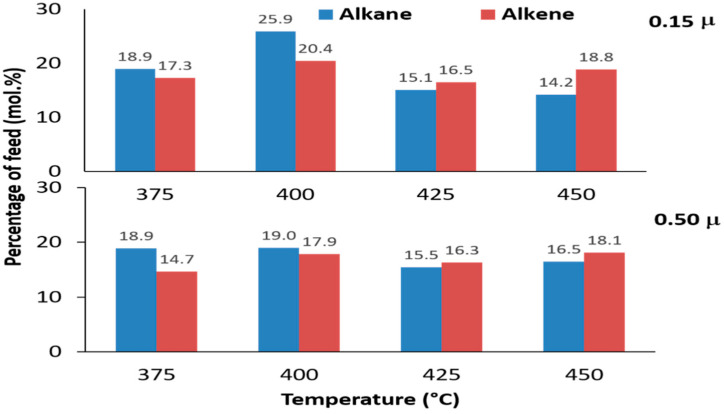
The n-alkane and 1-alkene composition liquid products from pyrolysis of mixed-metal basic soap at various reaction temperatures with two different Ca metal ratios (µ).

**Figure 5 molecules-27-00667-f005:**
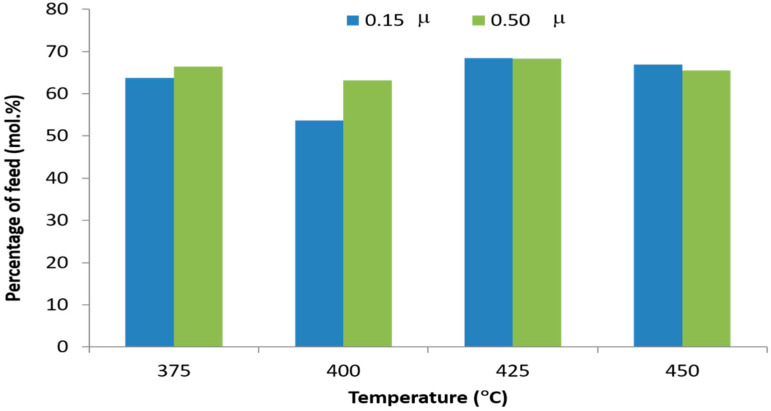
Iso-alkane composition of liquid products from pyrolysis of mixed-metal basic soap at various reaction temperature and with two different Ca/(Ca + Mg + Zn) metal ratios.

**Figure 6 molecules-27-00667-f006:**
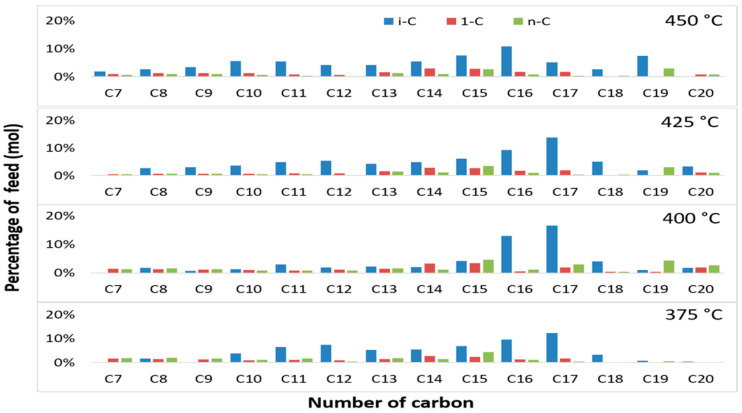
Compositions of hydrocarbons from pyrolysis of mixed metal basic soap at different reaction temperatures with Ca/(Ca + Mg + Zn) of 0.15 µ.

**Figure 7 molecules-27-00667-f007:**
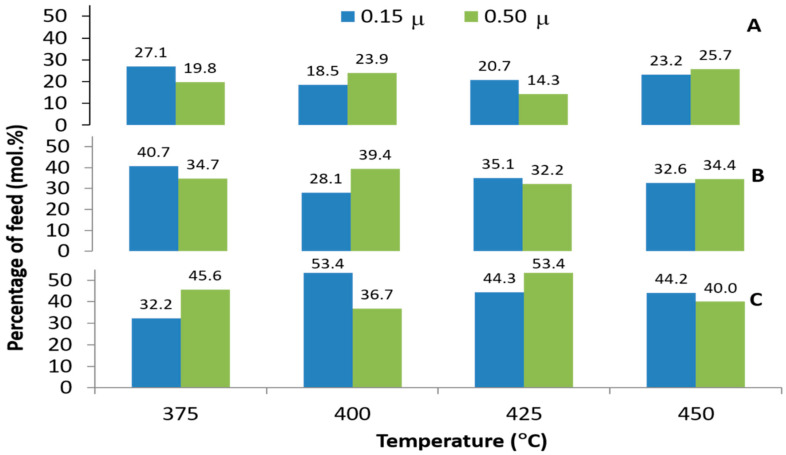
Gasoline (C_7_–C_11_)/(**A**), Avture (C_12_–C_15_)/(**B**) and diesel fraction (C_16_–C_19_)**/**(**C**) product composition of mixed metal basic soap pyrolysis at different reaction temperatures and two different Ca/(Ca + Mg + Zn) metal ratios.

**Table 1 molecules-27-00667-t001:** GC-MS analysis of PFAD and UFA.

Component	PFAD	UFA
tetradecanoic acid (C14:0)	1.7	3
hexadecanoic acid (C16:0)	40.2	17.6
9,12-octadecadienoic acid (C18:2)	5.2	5.4
9-octadecenoic acid (C18:1)	40.3	59.9
octadecanoic acid (C18:0)	7.8	5.3
methyl 18-methylnonadecanoate	1.3	--
Octadecanoic acid, 10-oxo-	--	1.9
Methyl 10D-hydroxyoctadecanoate	--	2.2
others	3.6	4.7

**Table 2 molecules-27-00667-t002:** Material balance of M-mix soap pyrolysis product.

Temperature (°C)	Yield of Product (wt.%)		
	Liquid Bio-Hydrocarbon	Solid Residues	Others (Include Water and Gas)
375	51.6	20.7	27.7
400	55.9	22.5	21.7
425	58.4	23.7	17.9
450	50.1	13.9	36.0

**Table 3 molecules-27-00667-t003:** Acid value of M-mix soap pyrolysis liquid product.

Temperature (°C)	Acid Value (mg KOH/100 g Sample)
375	0.66
450	0.39

**Table 4 molecules-27-00667-t004:** Gc-TCD analysis result of M-mix soap pyrolysis gas product.

Temperature (°C)	CO_2_	CH_4_	N_2_	O_2_
375	√	-	-	-
425	√	√	√	√
450	-	-	√	-

## Data Availability

Not applicable.

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
