# Peer review of "Sustainable Diesel from Pyrolysis of Unsaturated Fatty Acid Basic Soaps: The Effect of Temperature on Yield and Product Composition"

_molecules, 2022, doi:10.3390/molecules27030667_

Round 1
Reviewer 1 Report
Dear authors:
The topic and results presented in this work could be attractive to many potential readers, but the discussion and conclusion must be improved. Below are some comments to consider:
- All figures must be improved in quality. Consider the use of high-resolution images.
- For a better reader understanding of the logic of this work, consider showing the Methodology before to Results Section.
- Enrich the discussion of the data shown in tables and figures.
- Discuss in-depth the effect of temperature on the liquid-solid-gas yield, as well as the preferential formation of n-alkanes and i-alkanes in the fraction of liquid products. It would be appreciated to include a scheme of the reaction mechanisms that could explain the findings of this work.
- Please expand the Conclusion by introducing a fair comparison of the parameters (basic soap and fatty acids) discussing the fundamental origin of these differences.
Results:
- In Line 114, the text said: “…3200-3200 cm-1 [36]…” The interval that would correspond to the -OH groups would be 3200-3700 cm-1. Please check. Also, include the discussion of the other signals that appear in spectrum FTIR.
- In Line 127-129, the text said: “This can be explained as follows: at elevated temperatures from 245 to 450 °C, it has been expected that the cracking of bio hydrocarbons chains has already been obtained as a decarboxylation product”. This work only reported a temperature range of 375-450 °C, but the text said 245-450 °C. Please clarify. In case it refers to reported data, cite the source.
- In Line 131, the text refers to Figure 6. Try to follow the number sequence of figures according to their appearance in the manuscript.
- Figure 3. Axes and figures labels are missed.
- In Line 169, the text said: “Based on this mechanism, the yield of alkenes should be higher than alkanes, but not with this research.” Discuss in-depth the origin of these differences. It may be related to some catalytic effect due to species derived from Ca, Mg, and Zn hydroxides.
- In Line 170, the text said: “In this study, the amount of alkanes and alkenes at each temperature variation is almost 170 similar”, which contradicts what was noted in line 180: "The data from Figure 4 show that the maximum yield of alkene is reached at 400 °C 180 and 450 °C". Consider that Figure 4 shows no noticeable significant changes, although you should include the experimental error bars to determine significant differences.
- Figure 7. There is an inconsistency in the labels. Capital letters (A, B, and C) are used in the image, while lowercase letters (a, c, and c) are used in the caption.
- Line 262. Specify the temperature values.
Experimental:
- The sequence of subsections 3.2.X are fixed. Please correct.
- Specify the solvent used in the wash step in UFA preparation.
- What does the “B” refer to in "Burrette B"? in preparation of mix-metal hydroxides.
- Include units in the amount of formic acid added; what do you mean by (98-100%)? in preparation of basic soap.
- In line 327, the text fragment "…whit a capillary column (rtx-1)…" is repeated. Correct the wording of this statement.
- In line 330, the column and injector temperatures are 340 ° C?.
- Line 337: The terminology used (x:3.5) is confusing. Please correct.
Author Response
Response to the Referee’s comments
Response to Referee’s
Journal: Molecules
Manuscript ID:
Title: Sustainable Diesel from Pyrolysis of Unsaturated Fatty Acid Basic Soaps : The Effect of Temperature on Yield and Product Composition
Author(s):, Antonius Indarto, Endar Puspawiningtiyas *, Oki Muraza, Hary Devianto, Meiti Pratiwi, Subagjo Subagjo, Tirto Prakoso, Krisnawan Krisnawan, Usamah Zaki, Lidya Elizabeth, Yohanes Andre Situmorang, Tatang Hernas Soerawidjaja
Dear Editor,
Thank you for your useful comments and suggestions on our manuscript.
We have modified the manuscript accordingly, and detailed corrections are listed below:
The authors are appreciative that the referees have reviewed the paper and provided us with some valuable comments. Following are the changes which address the comments of the referees.
These adjustments have been made in the revised version.
The manuscript has been resubmitted to your journal. We look forward to your positive response.
Herewith, an itemized letter answering all the comments made and describing all changes made in response, or the reason why no change should be made. Below, we answer the comments one by one and we indicated clearly where changes have been made to the manuscript in response. Also, we submit our revised manuscript as a copy where the changes have been highlighted using a different color so that they can easily be identified.
Comments :
Reviewer 1 :
- All figures must be improved in quality. Consider the use of high-resolution images.
Authors’ response:
All figures have been changed with high-resolution images
- For a better reader understanding of the logic of this work, consider showing the Methodology before to Results Section.
Authors’ response:
Thank you for your suggestion, methodology has been described in section 2 with subtitle “Experiment” which consists of materials, preparation of unsaturated fatty acids, preparation of mixed metal hydroxides, Preparation of basic soap, Pyrolysis of basic soap, and measurement values and units. So that the reader’s understanding is better, the “experiment” subtitle has been changed with “Methodology” (line 96)
- Enrich the discussion of the data shown in tables and figures.
Authors’ response:
The discussion of the data shown in table and figures have been added (line 158-164, 188-195, 211-216, 241-246)
- Discuss in-depth the effect of temperature on the liquid-solid-gas yield, as well as the preferential formation of n-alkanes and i-alkanes in the fraction of liquid products. It would be appreciated to include a scheme of the reaction mechanisms that could explain the findings of this work.
Authors’ response:
Discuss in-depth the effect of temperature on the liquid-solid-gas yield has been added (line 188 – 195). The mechanism of formation of the isoalkane as dominant product in C17 has been added (line 299-310)
- Please expand the Conclusion by introducing a fair comparison of the parameters (basic soap and fatty acids) discussing the fundamental origin of these differences.
Authors’ response:
The sentences about introducing a fair comparison of the parameters (basic soap and fatty acids) discussing the fundamental origin of these differences have been added (line 366 & 369)
Result :
- In Line 114, the text said: “…3200-3200 cm-1 [36]…” The interval that would correspond to the -OH groups would be 3200-3700 cm-1. Please check. Also, include the discussion of the other signals that appear in spectrum FTIR.
Authors’ response:
Thank you for the correction, the text has been revised and the discussion of the other signal has been added (line 158-164)
- In Line 127-129, the text said: “This can be explained as follows: at elevated temperatures from 245 to 450 °C, it has been expected that the cracking of bio hydrocarbons chains has already been obtained as a decarboxylation product”. This work only reported a temperature range of 375-450 °C, but the text said 245-450 °C. Please clarify. In case it refers to reported data, cite the source.
Authors’ response:
Thank you for the correction, the text has been corrected and the source has been cited reference 27. (line 177)
- In Line 131, the text refers to Figure 6. Try to follow the number sequence of figures according to their appearance in the manuscript.
Authors’ response:
Thank you for the suggestion, it has been corrected following the number sequence of the figures.
- Figure 3. Axes and figures labels are missed.
Authors’ response:
Axes an figures labels have been completed
- In Line 169, the text said: “Based on this mechanism, the yield of alkenes should be higher than alkanes, but not with this research.” Discuss in-depth the origin of these differences. It may be related to some catalytic effect due to species derived from Ca, Mg, and Zn hydroxides.
Authors’ response:
The discuss in these dfferences have been added (line 241-246)
- In Line 170, the text said: “In this study, the amount of alkanes and alkenes at each temperature variation is almost 170 similar”, which contradicts what was noted in line 180: "The data from Figure 4 show that the maximum yield of alkene is reached at 400 °C 180 and 450 °C". Consider that Figure 4 shows no noticeable significant changes, although you should include the experimental error bars to determine significant differences.
Authors’ response:
Thank you for the correction, the contradicts sentence has been revised (line 256)
- Figure 7. There is an inconsistency in the labels. Capital letters (A, B, and C) are used in the image, while lowercase letters (a, c, and c) are used in the caption.
Authors’ response:
The labels in Figure 7 have been corrected, Thank you
Experimental :
- The sequence of subsections 3.2.X are fixed. Please correct.
Authors’ response:
I am sorry, I don’t understand what the mean. In section 3 includes 5 subsection yaitu :
3.1 Unsaturated fatty acids….
3.2 Identification of product…
3.3 Effects of temperature on the composition…
3.4 Effect of temperatureon iso-alkane…
3.5 Effects of temperature on yield…
- Specify the solvent used in the wash step in UFA preparation
Authors’ response:
The solvent has been specified (line 108)
- What does the “B” refer to in "Burrette B"? in preparation of mix-metal hydroxides.
Authors’ response:
Thank you for the very thorough correction, the sentence has been corrected (line 114)
- Include units in the amount of formic acid added; what do you mean by (98-100%)? in preparation of basic soap.
Authors’ response:
This means purity, the sentence has been fixed. (line 123)
- In line 327, the text fragment "…whit a capillary column (rtx-1)…" is repeated. Correct the wording of this statement.
Authors’ response:
The statement has been corrected (line 137)
- In line 330, the column and injector temperatures are 340 ° C?.
Authors’ response:
I have been checked the procedure and it is true that both use the same temperature, 340 °C (line 140)
- Line 337: The terminology used (x:3.5) is confusing. Please correct.
Authors’ response:
The terminology has been corrected
Reviewer 2 Report
This paper reported an interesting work on sustainable diesel production without hydrogen addition. The topic is important, and many results were obtained for analysis the production distribution. Before it be accepted in this journal, I suggest the issues should be considered.
- Line 113. More contents should be added into FTIR analysis of M-mix basic soap.
- Line 121. The valid digits should be consistent, and 58.4% is enough.
- Table 2. Is the unit %-wt? However, in the context, “%” is used in the whole manuscript.
- Table 3. How you calculate the acid value? The detailed method should be mentioned.
- Table 4. How you determine the gas production? Have you calculated the gas yield? Why there is an absence of 400 oC?
- Section 3.2.1. The schematic diagram of setup should be provided.
- The conclusion is not clear, the main findings and future direction should be mentioned.
- Some mistakes should be checked such as “ml” should be “mL”, “ZnCl2.4H2O” should be “ZnCl2•4H2O”, and “58.35 %” should be “58.35%”.
Author Response
Response to the Referee’s comments
Response to Referee’s
Journal: Molecules
Manuscript ID:
Title: Sustainable Diesel from Pyrolysis of Unsaturated Fatty Acid Basic Soaps : The Effect of Temperature on Yield and Product Composition
Author(s):, Antonius Indarto, Endar Puspawiningtiyas *, Oki Muraza, Hary Devianto, Meiti Pratiwi, Subagjo Subagjo, Tirto Prakoso, Krisnawan Krisnawan, Usamah Zaki, Lidya Elizabeth, Yohanes Andre Situmorang, Tatang Hernas Soerawidjaja
Dear Editor,
Thank you for your useful comments and suggestions on our manuscript.
We have modified the manuscript accordingly, and detailed corrections are listed below:
The authors are appreciative that the referees have reviewed the paper and provided us with some valuable comments. Following are the changes which address the comments of the referees.
These adjustments have been made in the revised version.
The manuscript has been resubmitted to your journal. We look forward to your positive response.
Herewith, an itemized letter answering all the comments made and describing all changes made in response, or the reason why no change should be made. Below, we answer the comments one by one and we indicated clearly where changes have been made to the manuscript in response. Also, we submit our revised manuscript as a copy where the changes have been highlighted using a different color so that they can easily be identified.
Comments :
Reviewer 2 :
- This paper reported an interesting work on sustainable diesel production without hydrogen addition. The topic is important, and many results were obtained for analysis the production distribution. Before it be accepted in this journal, I suggest the issues should be considered.
Authors’ response:
Thank you for the valuable comments
- Line 113. More contents should be added into FTIR analysis of M-mix basic soap.
Authors’ response:
Thank you for the suggestion. More contents have been added (line 158 – 164)
- Line 121. The valid digits should be consistent, and 58.4% is enough.
Authors’ response:
The digits have been corrected (line 171)
- Table 2. Is the unit %-wt? However, in the context, “%” is used in the whole manuscript.
Authors’ response:
Table 2 is correct using %-wt. Some “%” (Prosentase) have been added descripstion.
- Table 3. How you calculate the acid value? The detailed method should be mentioned.
Authors’ response:
Thank you for the suggestion, the acid value method has been mentioned (line 207-208)
- Table 4. How you determine the gas production? Have you calculated the gas yield? Why there is an absence of 400 oC?
Authors’ response:
The gas product data was obtained by calculating the difference between the pyrolyzed soap feed and the amount of liquid and solid products. While the water was formed very little so that the calculation is included with the gas product.
- Section 3.2.1. The schematic diagram of setup should be provided
Authors’ response:
The schematic diagram according to previous research. The citation has been added (line 133-134)
- The conclusion is not clear, the main findings and future direction should be mentioned.
Authors’ response:
The sentence has been added that shows the main finding (line 366-370)
- Some mistakes should be checked such as “ml” should be “mL”, “ZnCl2.4H2O” should be “ZnCl2•4H2O”, and “58.35 %” should be “58.35%”.
Authors’ response:
Thank you for the corrected. The mistakes have been revised. (line 122, line 163)

Round 2
Reviewer 1 Report
The comments were attended satisfactorily. Therefore, the current version of the manuscript can be considered for publication in this journal.